# Dynamic F-actin movement is essential for fertilization in *Arabidopsis thaliana*

**Tomokazu Kawashima[1†], Daisuke Maruyama[2,1], Murat Shagirov[3], Jing Li[1†], Yuki Hamamura[4], Ramesh Yelagandula[1†], Yusuke Toyama[5,1,3], Frédéric Berger[1,3]***

[1]Temasek Life Sciences Laboratory, National University of Singapore, Singapore, Singapore; [2]Nagoya Institute of Transformative Bio-Molecules, Nagoya University, Nagoya, Japan; [3]Department of Biological Sciences, National University of Singapore, Singapore, Singapore; [4]Division of Biological Sciences, Nagoya University Graduate School of Science, Nagoya, Japan; [5]Mechanobiology Institute, National University of Singapore, Singapore, Singapore

**Abstract** In animals, microtubules and centrosomes direct the migration of gamete pronuclei for fertilization. By contrast, flowering plants have lost essential components of the centrosome, raising the question of how flowering plants control gamete nuclei migration during fertilization. Here, we use *Arabidopsis thaliana* to document a novel mechanism that regulates F-actin dynamics in the female gametes and is essential for fertilization. Live imaging shows that F-actin structures assist the male nucleus during its migration towards the female nucleus. We identify a female gamete-specific Rho-GTPase that regulates F-actin dynamics and further show that actin–myosin interactions are also involved in male gamete nucleus migration. Genetic analyses and imaging indicate that microtubules are dispensable for migration and fusion of male and female gamete nuclei. The innovation of a novel actin-based mechanism of fertilization during plant evolution might account for the complete loss of the centrosome in flowering plants.

*For correspondence: fred@tll.org.sg

Present address: †Gregor Mendel Institute, Vienna, Austria

Competing interests: The authors declare that no competing interests exist.

## Introduction

Flowering plants have evolved a double fertilization process that requires the fusion of two sperm cells with the two female gametes, the egg cell and the central cell. This is achieved in a series of complex processes (*Kawashima and Berger, 2011*). Sperm cells are immotile and delivered to the female gametes by the pollen tube. The pollen tube is guided towards the female gametes (pollen tube guidance) and interacts with a synergid cell (two specialized cells associated with the female gametes) to release sperm cells (pollen tube reception) into the area proximal to the female gametes. After the fusion between the female and male gametes (plasmogamy), the male gamete nuclei migrate towards the female gamete nuclei and fuse (karyogamy), thus completing fertilization. Several factors and molecular mechanisms controlling these early events such as pollen tube guidance and reception have been recently characterized (*Higashiyama et al., 2001*; *Palanivelu et al., 2003*; *Kasahara et al., 2005*; *Escobar-Restrepo et al., 2007*; *Okuda et al., 2009*; *Kessler et al., 2010*; *Tsukamoto et al., 2010*; *Takeuchi and Higashiyama, 2012*; *Leydon et al., 2013*; *Liang et al., 2013*; *Ngo et al., 2014*), and mechanisms governing plasmogamy have been identified (*Mori et al., 2006*; *Von Besser et al., 2006*; *Mori et al., 2014*; *Sprunck et al., 2012*). However, how gamete nuclei migrate and fuse in flowering plants still remain largely elusive.

In most animals, microtubules organized by the centrosome control female pronucleus migration towards the male pronucleus (*Schatten, 1994*; *Reinsch and Gonczy, 1998*). By contrast, the essential components of the centrosome have been lost before the emergence of flowering plants (*Carvalho-Santos et al., 2011*), and the sperm cell nucleus moves towards the female gamete nucleus without

**eLife digest** Sexual reproduction involves combining the genetic material from two parents to create an offspring. The genetic material in the male sperm cell and the female egg cell is contained in the nucleus of each cell. Once these two cells fuse at fertilization, their nuclei must then navigate towards each other and fuse.

When an animal egg cell is fertilized, cable-like protein filaments called microtubules guide the two nuclei into contact. These microtubules are organized by a cellular structure called a centrosome. However, flowering plants do not have centrosomes; as such, it was unclear how genetic material from the sperm and egg cells is brought together after fertilization in flowering plants.

To investigate this, Kawashima et al. turned to a flowering plant commonly used in research, called *Arabidopsis thaliana*, and found that microtubules are not needed to guide the nuclei of the sperm and the egg cell after fertilization. Instead, another cable-forming protein—called F-actin—fulfills a similar role in Arabidopsis cells.

F-actin filaments often connect together to form a network; and when Kawashima et al. disrupted the F-actin in Arabidopsis egg cells, the nucleus of the sperm cell failed to fuse with that of the female. Pollen from Arabidopsis plants actually contains two sperm cells. One sperm cell fertilizes the egg cell; the other fertilizes the so-called 'central cell', which develops into a tissue that nourishes the plant embryo. Kawashima et al. found that the fertilization of both of these cells requires an intact F-actin network.

By looking more closely at F-actin networks in the larger central cell, Kawashima et al. discovered that the sperm nucleus becomes surrounded by a star-shaped structure of F-actin cables and that this F-actin structure migrates together with the sperm nucleus. The F-actin network constantly moves inward, from the edges of the cell towards the nucleus, prior to fertilization. This movement is essential for guiding the sperm nucleus towards the central cell nucleus.

Kawashima et al. also found that this continual movement of the F-actin network depends on a small signaling protein found in the central cell, called ROP8. It also involves a motor protein that normally transports "cargo", such as proteins and other molecules, inside cells by walking along the F-actin networks. However, rather than transporting the sperm nucleus as cargo, Kawashima et al. believe that the motor protein instead helps to maintain the inward movement of the F-actin network. One of the next challenges will be to investigate the molecular mechanism that underlies this motor protein's involvement in this dynamic F-actin network.

the participation of centrosomes in flowering plants. During plant evolution, the basal plants developed actin-based organelle movements (*Madison and Nebenfuhr, 2013*), which coincided with the centrosome loss in somatic cells in the basal land plants (*Vaughn and Harper, 1998*). However, the genomes of bryophytes, ferns, and most likely certain gymnosperms encode components of the centrosome and these organisms produce a centrosome specifically in spermatogenous cells (*Vaughn and Harper, 1998*). Indeed, microtubules are essential for gamete nuclear migration in the fern *Marsilea vestita* (*Kuligowski et al., 1985*), and these observations imply that gamete nuclear migration without the centrosome in flowering plants evolved separately from the actin-based organelle movement mechanism in somatic cells, leading to the question of how flowering plants control gamete nuclear migration without a centrosome. Immunofluorescence approaches revealed that corona structures of actin filaments around the sperm cells appear at the time of sperm cell release from the pollen tube prior to plasmogamy in many flowering plants (*Huang and Russell, 1994*; *Huang and Sheridan, 1998*; *Huang et al., 1999*; *Fu et al., 2000*; *Ye et al., 2002*). Changes in F-actin organization in the egg cell during fertilization are also evident (*Huang et al., 1999*; *Fu et al., 2000*), and indeed, an involvement of F-actin in gamete nuclear migration has been suggested during in vitro fertilization in rice (*Ohnishi et al., 2014*). Here, we report that in contrast to animals, microtubules are dispensable for fertilization and F-actin is the primary factor controlling sperm cell nucleus migration in *Arabidopsis*. Upon sperm nucleus entry in the central cell, F-actin generates an actin aster-like structure around the sperm cell nucleus while it migrates towards the female nucleus. The F-actin dynamic movement towards the central cell nucleus is established prior to fertilization and is controlled by a central cell-specific Rho-GTPase and myosin. Additional observations of F-actin function during egg

cell fertilization suggest similar conclusions. Hence, flowering plants innovated a novel actin-based mechanism of fertilization that made the centrosome dispensable.

## Results

### F-actin is essential for sperm nuclear migration in the egg cell

To investigate the requirement and function of F-actin during fertilization, we expressed the fluorescent reporter Lifeact-Venus (*Riedl et al., 2008*; *Era et al., 2009*) under the control of the egg cell-specific *EC1* promoter (*Sprunck et al., 2012*) to visualize the actin cytoskeleton in the egg cell (*Figure 1A,B*). Lifeact-Venus marked cables were disassembled after treatment with the actin polymerizing inhibitor Latrunculin A (LatA; *Figure 1C*). Pharmacological analysis by applying inhibitor drugs is useful to dissect out the cytoskeleton function at the cellular level. However, treatment with actin polymerization inhibitors disrupts functions in all cells when applied to tissues such as ovules and thus prevents the analysis on specific cytoskeleton functions in a specific cell-type. To overcome this problem, the semi-dominant negative *ACTIN* transgene (*DN-ACTIN*; *Kato et al., 2010*) was introduced to disrupt F-actin specifically in female gametes. DN-ACTIN contains one amino acid substitution in the hydrophobic loop of *Arabidopsis* ACT8, which causes instability and fragmentation of actin filaments, leading to incomplete yet strong disruption of actin cytoskeleton (*Kato et al., 2010*). Consistent with the effect of DN-ACTIN reported previously, the filamentous structures shown in the wild-type (WT) egg cell became much shorter and generated aggregates in the egg cell expressing DN-ACTIN (*Figure 1D*). In WT plants, fertilization leads to karyogamy followed by decondensation of the chromatin from the male nucleus (*Figure 1E*; *Ingouff et al., 2007*). Egg cell fertilization initiates embryo development while the fusion of the other sperm cell with the central cell leads to endosperm development (*Figure 1A,F*). By contrast, fertilization of the egg cell expressing DN-ACTIN failed as the sperm nucleus did not fuse with the egg cell nucleus and the sperm chromatin remained condensed (*Figure 1G*; Line 1, 35% defects in *DN-ACTIN* [n = 104] compared to 0% defects in WT [n = 98]). Karyogamy was

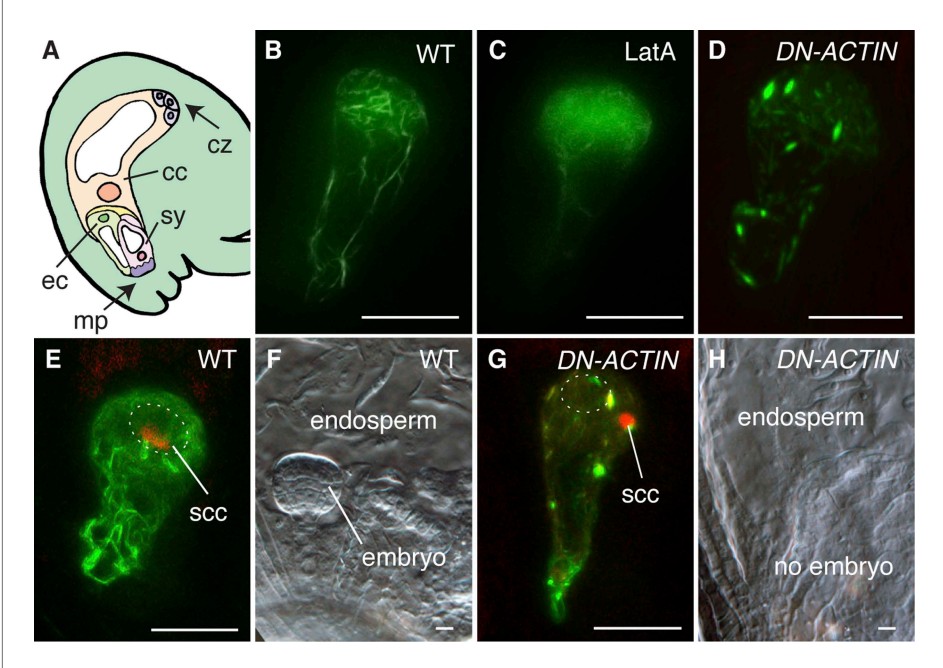

**Figure 1**. F-actin is required for egg cell fertilization. (**A**) Cartoon of *Arabidopsis* mature ovule. cc, central cell; cz, chalaza; ec, egg cell; mp, micropyle; sy, synergid. (**B–D**) Egg cell actin cables (**B**) become disassembled in LatA treatment (**C**) and in *DN-ACTIN* (**D**). (**E** and **F**) Successful fertilization marked by decondensation of the sperm cell chromatin (ssc, red) into the egg cell nucleus (dashed oval) (**E**), resulting in a normal embryo in WT (**F**). (**G** and **H**) Egg cell expressing DN-ACTIN shows arrests in sperm cell nuclear migration (**G**) and embryo development (**H**). Scale bar = 10 μm.

prevented only in the egg cell expressing DN-ACTIN but not in the central cell, resulting in a seed containing endosperm without an embryo [*Figure 1H*; Line 1, 27% defects in *DN-ACTIN* (n = 110) compared to 0% defects in WT (n = 389)]. Taken together, these results suggest that actin cytoskeletons are required for egg cell fertilization. Consistently, other independent transgenic lines showed similar seed developmental arrest [Line 2, 20% defects (n = 125); Line 3, 22% defects (n = 114)]. Not all ovules of DN-ACTIN expressing lines showed the fertilization defect, likely because a certain fraction of actin filaments was still functional.

## F-actin associates with the sperm nucleus and is essential for sperm cell nucleus migration in the central cell

We further investigated the requirement of F-actin during fertilization in the central cell that is about five times larger than the egg cell, thus enabling more detailed F-actin dynamics visualization. We expressed Lifeact-Venus under the control of the central cell-specific *FWA* promoter (*Kinoshita et al., 2004*). F-actin in the WT mature central cell showed that F-actin is organized in three major distinct structures: (i) a ring-shaped actin network at the micropylar end of the central cell surrounding the synergid cells and the egg cell; (ii) long actin cables at the periphery of the central cell, extending from the chalazal region to the boundary with the egg cell; and (iii) a track formed of a bundle of actin cables, running from the micropylar actin ring to the central cell nucleus (*Figure 2A*; *Figure 2—figure supplement 1*). As expected, these structures disassembled in response to LatA treatment (*Figure 2—figure supplement 2*). During fertilization, the track of actin cables became associated with the migrating sperm cell nucleus in the central cell from the site of fusion between the sperm cell and the central cell to the central cell nucleus (*Figure 2B,C*). Time-lapse image analysis further confirmed F-actin association to the sperm cell nucleus during its migration in the central cell (*Figure 3*; *Video 1*). Upon completion of karyogamy, the micropylar actin ring and bundles of F-actin associated with the sperm cell nucleus disassembled (*Figure 2D*; *Video 2*), suggesting that F-actin assembles transient dynamic structures dedicated for fertilization. The constitutive expression of DN-ACTIN in the central cell

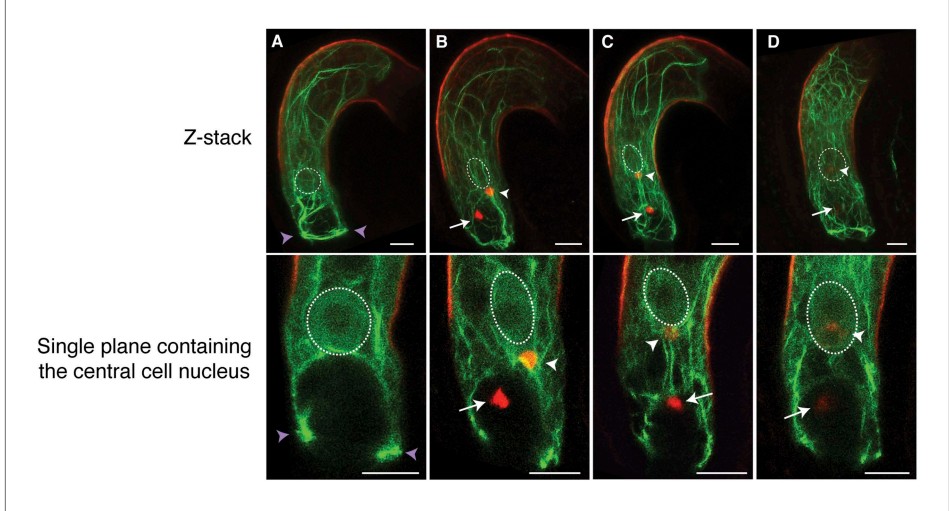

**Figure 2**. F-actin in the central cell associates with the sperm cell nucleus during migration. (**A**–**D**) Z-Stacked (top) and single plane (bottom) images of F-actin in the WT central cell during fertilization. The central cell F-actin before fertilization (**A**), at the onset (**B**), and completion (**C**) of nuclear migration, and karyogamy (**D**) were shown. White arrows and arrowheads point to sperm chromatins in the egg and central cells, respectively. White-dashed ovals display the position of the central cell nucleus. Purple arrowheads point to the F-actin ring structure present at the micropylar end of the central cell. Scale bar = 10 µm.

The following figure supplements are available for figure 2:

**Figure supplement 1**. Unique F-actin structures in the mature central cell before fertilization.

**Figure supplement 2**. Effects of LatA and DN-ACTIN on F-actin and fertilization in the central cell.

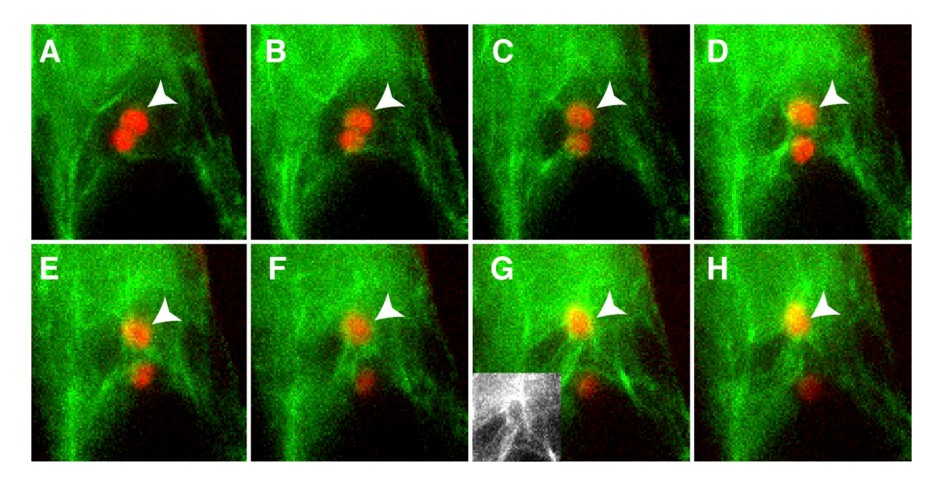

**Figure 3**. Actin cables generate an aster-like structure around the sperm cell nucleus during migration in the central cell. (**A–H**) 1-min interval time-lapse image montage during sperm cell nuclear migration. After gamete fusion, the sperm cell nuclei (red ovals) start moving towards opposite directions (**A–C**). In the central cell, actin cables (green) become associated with the sperm cell nucleus (arrowhead) after its entry (**D** and **E**) and assemble an aster-like structure around the sperm cell nucleus (**F–H**). Inlet in panel **G** shows actin cables (white) around the sperm cell nucleus in the central cell without sperm chromatin fluorescence. See also *Video 1*.

prevented fertilization [*Figure 2—figure supplement 2*; Line 1, 30% defects in *DN-ACTIN* (n = 120) compared to 0% defects in WT (n = 153)] and endosperm development [*Figure 2—figure supplement 2*; Line 1, 25% defects in *DN-ACTIN* (n = 412) compared to 0% defects in WT (n = 233)]. Other independent transgenic lines expressing DN-ACTIN in the central cell also showed similar rates of endosperm developmental defect (Line 2, 22% defects [n = 300]; Line 3, 21% defects [n = 434]). Occasionally, a few nuclear divisions were observed in the endosperm (1–2%), resembling a phenotype described in ovules fertilized by *cdka;1* sperm cells that are unable to undergo karyogamy (*Aw et al., 2010*), thus supporting further that sperm cell fusion triggers central cell division. Taken together, these results suggest that F-actin also plays an important role in fertilization of the central cell.

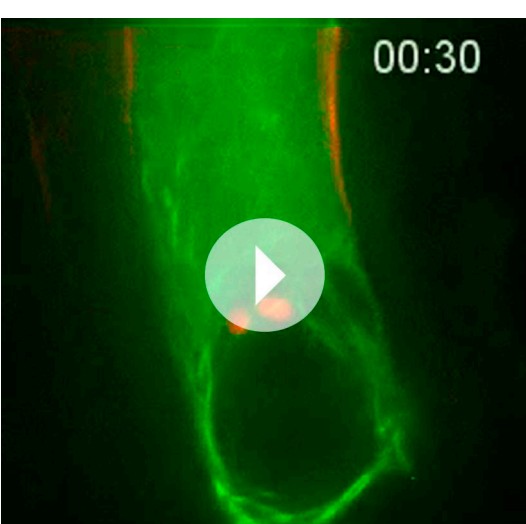

**Video 1**. Time-lapse image of F-actin dynamics and sperm cell nuclear migration in the WT central cell. Sperm cell chromatins (red ovals) are marked by *pH3.10::H3.10-mRFP1* and F-actin in the central cell (green) is marked by *pFWA::Lifeact-Venus*. Autofluorescence marks the central cell outline (red line). Images were taken at 30 s intervals.

## Inducible disruption of F-actin to investigate its involvement in gamete nuclear migration

*Arabidopsis* female gametes undergo a maturation phase once they become specified (*Drews and Koltunow, 2011*), and the *FWA* promoter is active soon after the central cell is determined. Therefore, we could not rule out the possibility that the constitutive expression of DN-ACTIN in the central cell by the *FWA* promoter might impair the maturation of the central cell, which in turn would be responsible for fertilization failure. To overcome this problem, we designed a dexamethasone (DEX)-inducible expression system (*Samalova et al., 2005*) to activate *DN-ACTIN* after central cell maturation (*Figure 4A,B*). As the

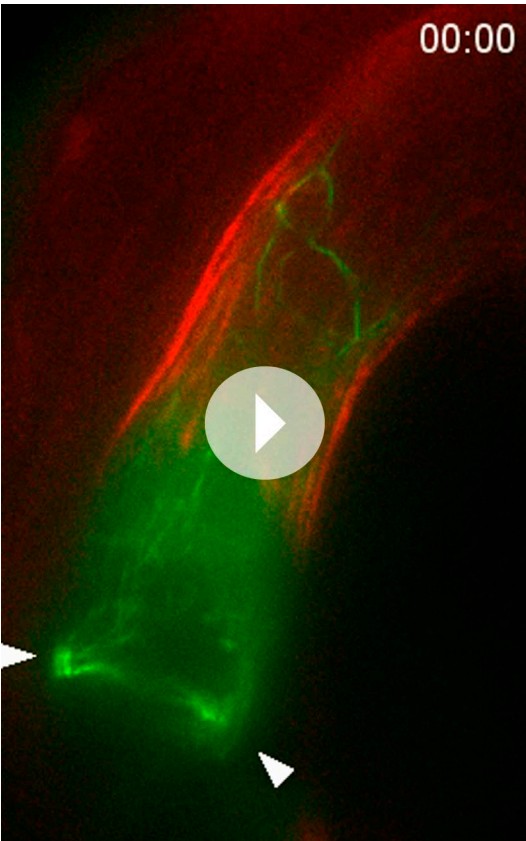

00:00

**Video 2**. Time-lapse image of F-actin micropylar ring-like structure in the central cell during fertilization. Sperm cell chromatins (red ovals) are marked by *pH3.10::H3.10-mRFP1* and F-actin in the central cell (green) is marked by *pFWA::Lifeact-Venus*. Arrowheads point the micropylar F-actin ring in the central cell. Autofluorescence marks the central cell outline (red line). Images were taken at 1-min intervals.

activator construct, the LhGR, constituted from the steroid-binding domain of a rat glucocorticoid receptor (GR, residues 508–795), the mutated DNA-binding *lac* repressor (lacl[His17], residues 1–330), and yeast *GAL4* transcription activation domain II (Gal4-II, residues 768–881), was flanked by the *FWA* promoter. The reporter construct consists of the nuclear-localized mCherry fluorescent protein (NLS-mCherry) and *DN-ACTIN* genes, connected by a self-cleaving 2A peptide derived from porcine teschovirus-1 (P2A), and this transgene was flanked by the four tandem repeats of the *lac* operator with the *cauliflower mosaic virus 35S* minimal promoter (TATA; *Figure 4A*). During translation, the P2A sequence causes ribosome skipping, generating two individual proteins from a single transgene (*El Amrani et al., 2004*; *Kim et al., 2011*). As a result, two proteins, NLS-mCherry and DN-ACTIN, are produced in the central cell by DEX application (*Figure 4B*), and the NLS-mCherry can be used as a marker for spatio-temporal DEX induction. Consistent with the result from the constitutive expression of DN-ACTIN by the *FWA* promoter, DN-ACTIN induced by DEX in the mature central cell disrupted F-actin organization and prevented sperm cell nucleus migration towards the female nucleus and karyogamy (*Figure 4C,E*). Endosperm development was also prevented in DEX-treated plants (*Figure 4D,F*). These results confirmed that F-actin is essential for the migration of the sperm cell nucleus to the central cell nucleus.

## Female gamete-specific Rho-GTPase is responsible for F-actin dynamics in the central cell

We observed that, upon entry into the central cell, the sperm cell nucleus becomes surrounded by an increasing number of F-actin cables (*Figure 3A–D*), followed by the formation of a dynamic aster-like structure around the sperm cell nucleus (*Figure 3E–H*). This aster-like structure of F-actin was only observed around the sperm cell nucleus in the central cell after plasmogamy (100%, n = 48) and was never observed in unfertilized central cells (0%, n = 72). Consistently, the time-lapse imaging showed the presence of aster-like F-actin structure only during sperm cell nucleus migration [100% (n = 5); 0% in unfertilized ovule (n = 14)].The F-actin aster-like structure migrated together with the sperm cell nucleus towards the central cell nucleus (*Video 1*), suggesting the involvement of specific actin regulators in this process. In plants, F-actin assembly is regulated by formins and ARP2/3 complex under the control of plant-specific Rho-GTPases (ROP; *Craddock et al., 2012*; *Henty-Ridilla et al., 2013*). Transcriptional profiling (*Borges et al., 2008*; *Le et al., 2010*; *Wuest et al., 2010*; *Belmonte et al., 2013*) suggested that *ROP8* is specifically active in the central cell and the endosperm (*Figure 5A*). Indeed, we found that the *ROP8* promoter is specifically active in the central cell (*Figure 5B*) and the fusion protein Venus-ROP8 associates to the plasma membrane of the central cell (*Figure 5C*). Consistently, the ectopic expression of GFP-ROP8 also showed its association to the plasma membrane in root epidermal cells (*Figure 5—figure supplement 1*), indicating that ROP8 sub-localization is controlled by general machinery common between the gametophytic central cell and somatic root epidermal cell. To understand the function of ROP8 in the central cell during fertilization, we investigated F-actin organization in central cells expressing constitutively active

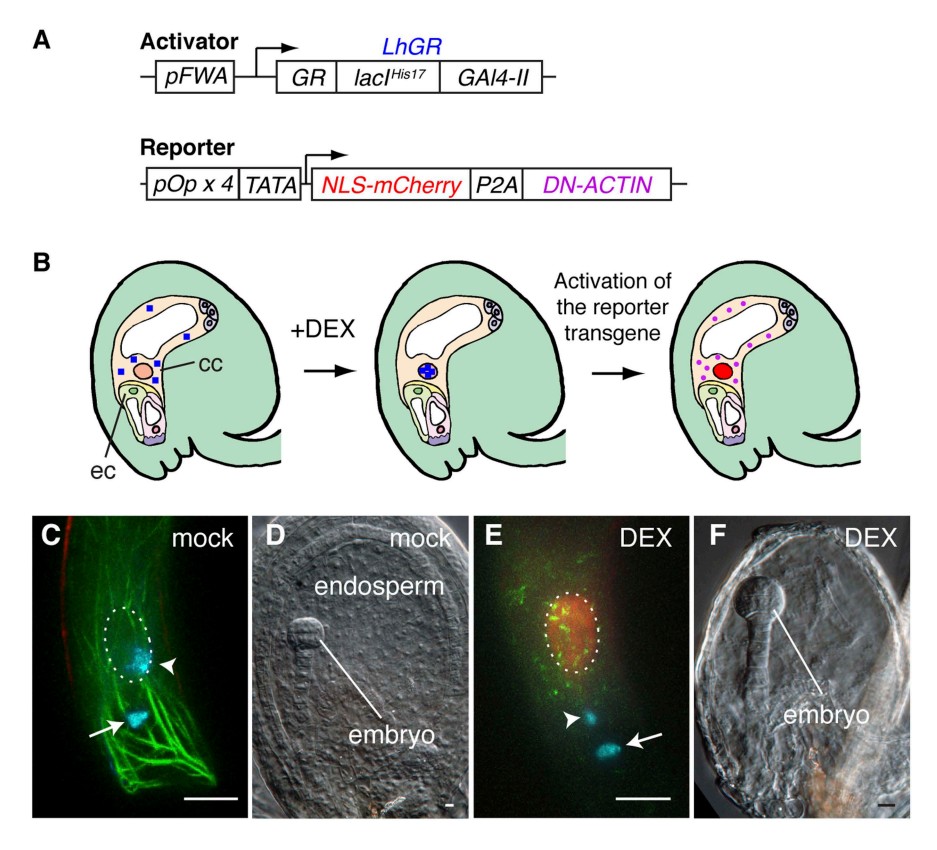

**Figure 4**. Spatio-temporal control of F-actin disruption demonstrates importance of F-actin for fertilization.
(**A**) Schematic representations of the transcription activator (top) and reporter (bottom) constructs of DEX inducible expression strategy. The transcription activator LhGR is placed under the control of the *FWA* promoter. The reporter construct consists of *NLS-mCherry* and *DN-ACTIN* genes, connected by a self-cleaving 2A peptide derived from porcine teschovirus-1 (P2A), flanked by the four *lac* operators with the *CaMV 35S* minimal promoter (TATA). (**B**) Schematic representations of *DN-ACTIN* induction in the mature central cell by DEX. When DEX is applied, LhGR (blue square) localizes to the central cell nucleus. As a result, the reporter transgene becomes activated specifically in the central cell. During translation, the P2A peptide is recognized for the ribosomal skip, resulting in two separate proteins: NLS-mCherry and DN-ACTIN. NLS-mCherry proteins are transferred to the nucleus (red nucleus), and DN-ACTIN (purple ovals) functions in the same cell for F-actin disruption. cc, central cell; eg, egg cell. (**C**–**F**) *DN-ACTIN* induction in the mature central cell disrupts nuclear migration and endosperm development. Mock treatment showing sperm chromatin decondensation (cyan) in both the egg and central cells (**C**), leading to normal embryo and endosperm development (**D**). Central cell *DN-ACTIN* induction marks specifically the central cell nucleus (dashed oval) and causes F-actin disruption (**E**), preventing sperm cell nuclear migration (arrowhead) while karyogamy takes place in the egg cell (arrow), resulting in embryo but no endosperm development (**F**). Scale bar = 10 μm.

and dominant-negative forms (*Yang, 2002*) of ROP8 (*CA-ROP8* and *DN-ROP8*, respectively) under the control of the *ROP8* promoter. Although F-actin structures in both *CA-ROP8* and *DN-ROP8* appeared similar to WT (*Figure 5—figure supplement 2*), only *DN-ROP8* lines showed that the sperm cell chromatin in the central cell remained condensed and unfused with the central cell nucleus, when the sperm cell chromatin had already fused with the egg cell nucleus and appeared decondensed (*Figure 5D*; Line 1, 42% defects [n = 98], Line 2, 36% defects [n = 102]). This phenotype was similar to the fertilization defect observed in central cells expressing DN-ACTIN (*Figure 4*).

We measured the dynamics of F-actin around the central cell nucleus, where the sperm cell nucleus migration occurs, and observed a predominant F-actin movement from the cell periphery towards the nucleus (*Video 3*), hereafter referred to as inward movement (*Figure 6A,B*). By contrast,

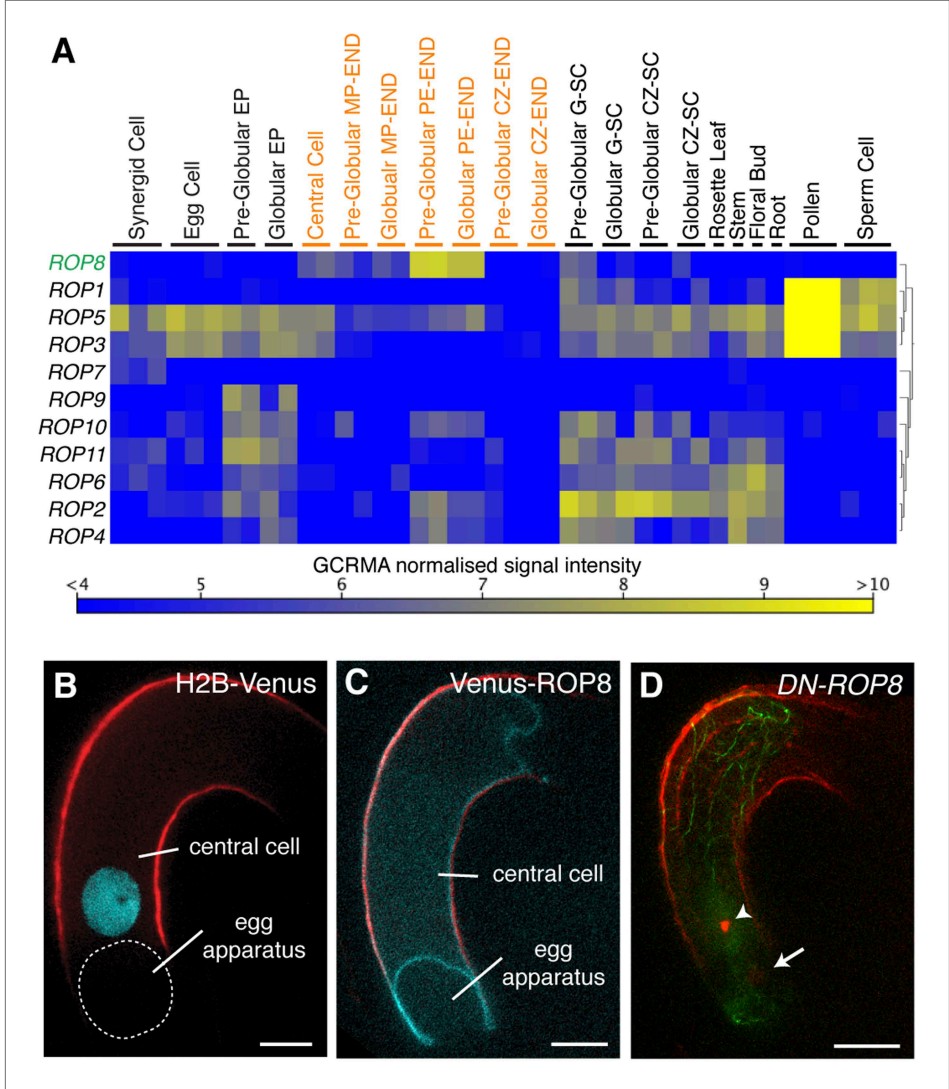

**Figure 5**. ROP8 is specific to the central cell and involved in fertilization. (**A**) *Arabidopsis ROP* encoding genes expression heat map constructed from public data. CZ-END, chalazal endosperm; CZ-SC, chalazal seed coat; EP, embryo proper; G-SC, general seed coat; MP-END, micropylar endosperm; PE-END, peripheral endosperm. (**B** and **C**) H2B-Venus (**B**) and Venus-ROP8 (**C**) under the *ROP8* promoter show expression specifically in the central cell and its association to the plasma membrane (cyan), respectively. Autofluorescence marks the central cell outline (red line). (**D**) *DN-ROP8* shows central cell fertilization defect. Sperm chromatin in the egg cell (arrow) is already decondensed, but sperm chromatin remains condensed in the central cell (arrowhead). Scale bar = 10 μm.

The following figure supplements are available for figure 5:

**Figure supplement 1**. Ectopic expression of ROP8 in the root also shows its association to the plasma membrane.

**Figure supplement 2**. The expression of neither *constitutively-active ROP8* (*CA-ROP8*) nor *dominant-negative ROP8* (*DN-ROP8*) under the control of the *ROP8* promoter affect the F-actin structure (green) in the central cell.

F-actin movement was extremely reduced in central cells expressing DN-ROP8 (*Figure 6C–E*; *Videos 3*; *Video 4*; *Video 5*; 45% and 30% defects in *DN-ROP8* Line 1 [n = 11] and Line 2 [n = 10], respectively, compared to 0% defect in WT [n = 7] and *CA-ROP8* Line 1 [n = 5] and Line 2 [n = 6]). Taken together, these results show that ROP8 controls novel constant inward dynamics of F-actin movement in the central cell and that this ROP8-dependent inward F-actin movement is required

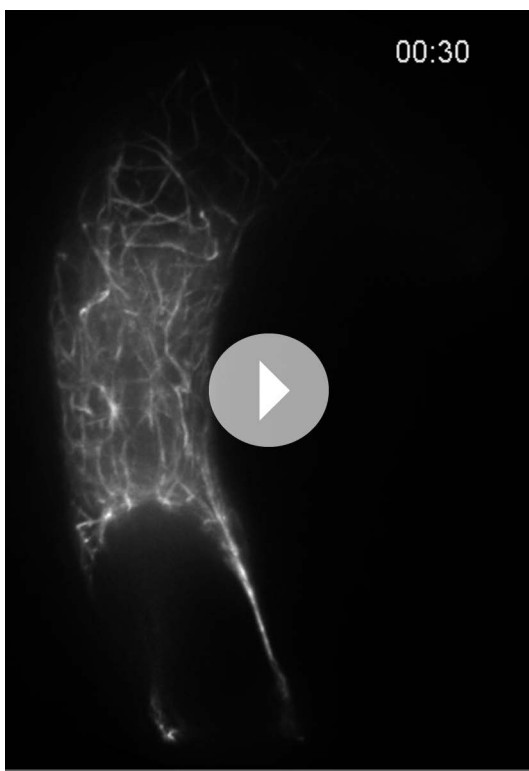

00:30

**Video 3**. Time-lapse image of F-actin dynamics in the mature WT central cell. F-actin in the central cell (white) were marked by *pFWA::Lifeact-Venus*. Images were taken at 30-s intervals.

for the proper transport of the sperm cell nucleus to the central cell nucleus. Although F-actin flows towards the central cell nucleus, we did not detect accumulation of F-actin around the nucleus (*Video 3*), suggesting that F-actin disassembly takes place around the central cell nucleus.

## Dynamic F-actin movement in the central cell is myosin dependent

In plants, the motor protein myosin generates force to transport organelles along actin filaments within a cell (*Madison and Nebenfuhr, 2013*), including the nuclear movement in somatic cells (*Tamura et al., 2013*). To investigate an involvement of myosin, we applied the myosin inhibitors 2,3-butanedione monoxime (BDM) and N-ethylmaleimide (NEM; *Hoffmann and Nebenfuhr, 2004*) to dissected ovules containing the mature central cell. In the presence of either myosin inhibitor, F-actin movement significantly reduced as observed in central cells expressing*DN-ROP8* (*Figure 6C–E*; *Videos 6*; *Video 7*; 100% defect with BDM [n = 11] and NEM [n = 7], compared to 0% defect in mock treated ovules [n = 7 and 5, respectively]). These results indicate that myosin plays a role in the dynamics of F-actin in the central cell and contributes to a net inward F-actin movement.

## Microtubules are dispensable for male gamete nuclear migration

Although F-actin appears to be the main factor controlling male gamete nuclear migration, it is still possible that microtubules are also involved in male gamete nuclear migration in flowering plants even in the absence of a centrosome. To investigate the requirement of microtubules in *Arabidopsis* fertilization, we used a null mutant allele of *PORCINO* (*POR*) that encodes a subunit of TUBULIN FOLDING COFACTOR C (*Steinborn et al., 2002*). Self-pollinated *por/+* heterozygous plants produce about 25% defective embryos that do not assemble microtubule and abort, indicating that viable *por* gametes fuse to produce *por/por* homozygous seeds that are not viable (*Steinborn et al., 2002*). We expressed the TagRFP-TUBULIN ALPHA-5 in the central cell to visualize microtubules in *por/+* plants that produce 50% *por* and 50% WT gametes. In contrast to WT, about half of the female gametes from *por/+* mutants did not show microtubule structures (*Figure 7A,B,F*). This proportion was consistent with the requirement of POR function for microtubule assembly. WT and *por* mutant female gametes fertilized with either WT or *por* sperm cells showed successful karyogamy leading to the decondensation of chromatin from the sperm cell nucleus (*Figure 7C,D*; 97% in WT [n = 102] and 96% in *por/+* [n = 96]). Taken together, these results show that fertilization takes place despite defective microtubule assembly and as a result, *por/por* embryo and endosperm development initiates. Mutations in genes encoding the other subunits (A, D and E) of TUBULIN FOLDING COFACTOR cause a seed abortion phenotype similar to that described in *por* (*Steinborn et al., 2002*), further demonstrating that microtubules are not absolutely required for fertilization in *Arabidopsis*. Additionally, we investigated whether F-actin and microtubules interact to coordinate nuclear migration. However, microtubule organization was not affected in central cells expressing DN-ACTIN. Conversely in *por* central cells, the actin cytoskeleton retained its organization similar to WT (*Figure 7E,F*). These observations support the idea that the actin cytoskeleton is the primary factor controlling sperm cell nuclear migration during fertilization in *Arabidopsis* in contrast to most animal species in which microtubules play a major role in pronuclear migration.

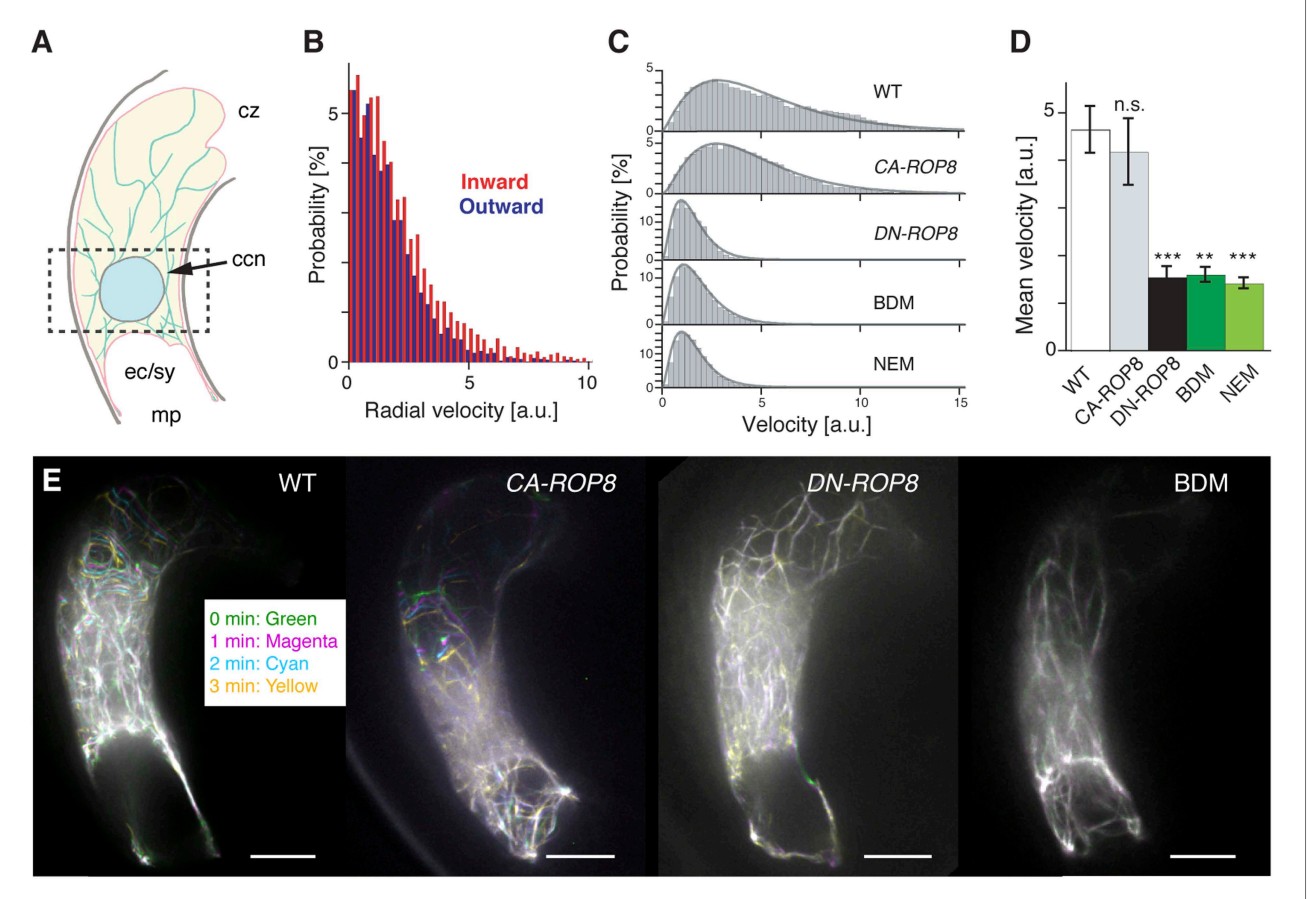

**Figure 6**. Controls of F-actin dynamics in the central cell. (**A**) A scheme of the mature *Arabidopsis* ovule. F-actin (light green cables) velocities of the entire central cell and around the nucleus (dashed box) were analyzed. ccn, central cell nucleus; cz, chalaza; ec/sy, egg/synergid cells complex; mp, micropyle. (**B**) F-actin velocity around the central cell nucleus. Inward and outward represent F-actin movement direction towards and away from the nucleus, respectively. a.u., arbitral unit. (**C**) The representative probability distributions of velocities integrated over the entire central cell area and time. Solid lines represent the fitted gamma distribution. (**D**) The mean velocities of central cell F-actin movement. Error bars represent the standard deviation of five to seven biological replicates. n.s., not significant; ***, p < 0.001; **, p < 0.01, student t-test. (**E**) Stacks of central cell F-actin time-lapse. F-actin movements are marked by rainbow colors while white color results from overlapping multiple colors, indicating less movement. Scale bar = 10 μm.

## Discussion

Nuclear movement in both gamete and somatic cells has been extensively studied in animals. Pronuclear migration in the fertilized egg is microtubule dependent, but both actin filament and microtubules are indispensable for nuclear positioning in many somatic cells (*Reinsch and Gonczy, 1998*; *Gundersen and Worman, 2013*). In plants, nuclear positioning during cell division and photo-relocation as well as tip growth of root hair and pollen tube has been investigated (*Takagi et al., 2011*; *Higa et al., 2014*), and actin filaments play a critical role in these nuclear positioning events in plant somatic cells. By contrast, very little was known about gamete nuclear migration in plants. During evolution, the centrosome loss in spermatogenous cells of land plants did not coincide with either the centrosome loss in somatic cells or the initiation of the actin-based organelle movement system in plants, indicating that gamete nuclear migration in flowering plants might depend on a distinct mechanism. Here, we identified that the dynamic F-actin movement towards the central cell nucleus from the cell periphery constantly occurs prior to the sperm cell nucleus entry, and this F-actin movement is essential for sperm cell nucleus migration in *Arabidopsis*. Upon entry into the central cell, the sperm nucleus becomes surrounded by F-actin, generating an aster-like structure that migrates together with the sperm cell nucleus to join the central cell nucleus for karyogamy. We also found that this dynamic F-actin movement in the central cell is controlled by gamete-specific Rho-GTPase ROP8 and is

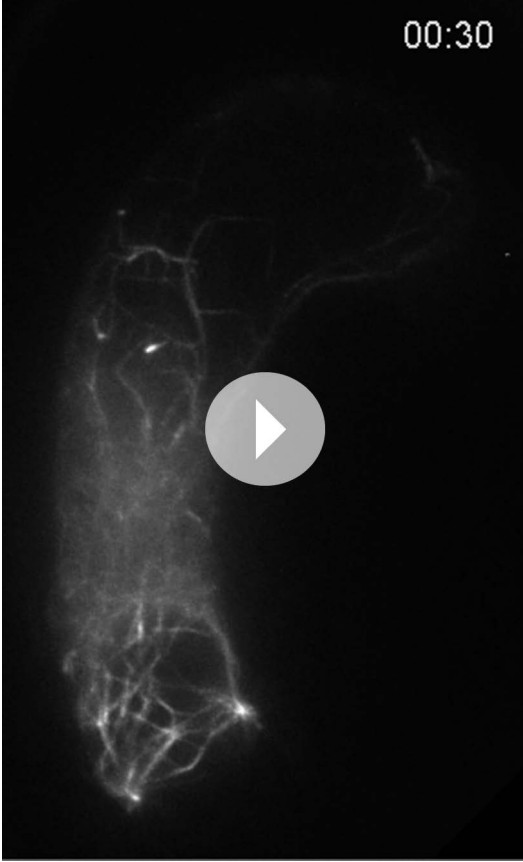

00:30

**Video 4**. Time-lapse image of F-actin dynamics in the mature *CA-ROP8* central cell. F-actin in the central cell (white) was marked by *pFWA::Lifeact-Venus*. Images were taken at 30-s intervals.

myosin dependent. On the contrary, microtubules are dispensable for sperm cell nuclear migration in both female gametes. Taken together, our results show that the sperm cell nuclear migration in flowering plants is controlled by a novel actin-based mechanism that generates a constant F-actin inward movement towards the central cell nucleus. The dynamic movement of F-actin towards the central nucleus, F-actin aster-like structure around the sperm cell nucleus during migration, and precise spatio-temporal disruption of F-actin after central cell maturation by the DEX system suggest direct involvement of F-actin for sperm cell nuclear migration. However, it still remains to be determined whether F-actin directly interacts with the sperm cell nucleus for migration. Further investigations should be made to rule out the possibility that the sperm cell nuclear migration defect by DN-ACTIN expression is not due to general perturbation of cellular structure.

We also describe that F-actin is organized as unique structures in the central cell. In addition to the F-actin aster-like structure and the F-actin track that associate with the sperm cell nucleus migration, the central cell features a ring-shaped actin network at the micropylar end that surrounds the egg apparatus (the synergid cells and the egg cell). The F-actin ring might maintain the position of the egg apparatus for fertilization or it might act as a clamp to maintain mechanical constraints that are required for efficient sperm cell delivery. Upon completion of karyogamy, all three organized F-actin structures become disassembled, suggesting that these unique actin structures are specialized to achieve fertilization.

Rho-GTPases are well-studied signaling factors controlling cytoskeleton dynamics (*Yang, 2002*). In the WT central cell, the constant inward movement of F-actin from the cell periphery to the nucleus occurs prior to fertilization, and this constant inward movement becomes significantly reduced in *DN-ROP8*, but not in *CA-ROP8* transgenic lines. These results suggest that ROP8 might be constantly active in the entire central cell plasma membrane, generating a constitutive dynamic inward movement of F-actin. This could explain why *DN-ROP8*, but not *CA-ROP8* transgenic lines affected F-actin dynamics significantly (*Figure 6*). In general, the activation/inactivation of ROPs is controlled spatially at the plasma membrane, generating subdomains within a cell for downstream events such as secondary cell wall formation (*Oda and Fukuda, 2012*) and pavement cell indentation (*Xu et al., 2010*). Although the machinery by which ROP8 is associated to the plasma membrane is general and the activity of ROP8 in the central cell should be investigated further, this putative constant active mode of ROP8 in the central cell might shed light on a new facet of Rho-GTPase function in plant cells. F-actin function is also important for sperm cell nuclear migration in the *Arabidopsis* egg cell described here and in the rice egg cell shown by in vitro fertilization assay (*Ohnishi et al., 2014*). It is plausible that a distinct ROP plays a similar role during egg cell fertilization, but it is also possible that other mechanisms control F-actin-dependent migration of the male nucleus to the egg cell nucleus.

According to observations in somatic cells (*Tamura et al., 2013*), myosin was expected to tether the nucleus and enable its migration as a cargo. However, the impairment of F-actin movement in the central cell halts sperm cell nuclear migration even in the presence of the WT-like structure of actin filaments shown in *DN-ROP8*. Hence, myosin might not transport the sperm cell nucleus

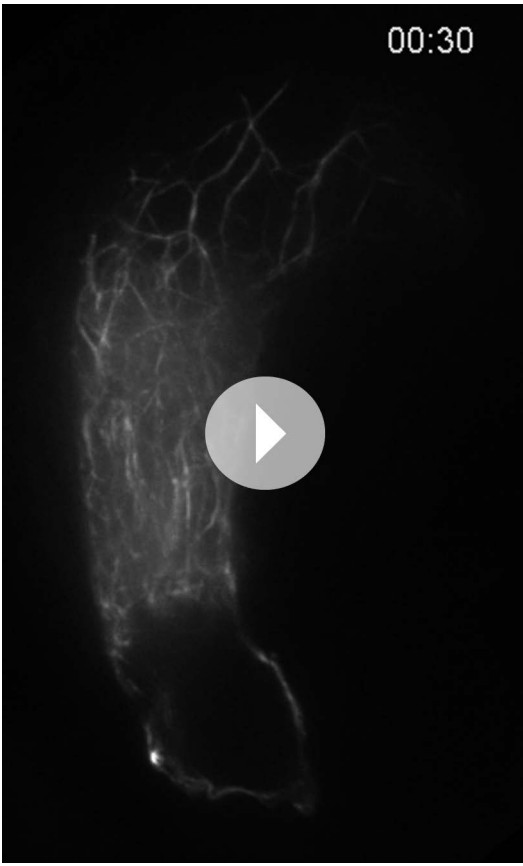

**Video 5**. Time-lapse image of F-actin dynamics in the mature DN-ROP8 central cell. F-actin in the central cell (white) is marked by *pFWA::Lifeact-Venus*. Images were taken at 30-s intervals.

towards the female nucleus as a cargo. Our observations rather suggest that myosin affects F-actin dynamics directly. In vitro, myosin also creates bipolar filaments and cross-links two actin filaments, generating a non equilibrium dynamic F-actin movement (*Mizuno et al., 2007*). This type of dynamics is similar to the inward F-actin movement observed in the central cell, suggesting that myosin is directly involved in the dynamic inward movement of F-actin in this cell type. Yet, we have not identified which specific myosin is involved and detailed myosin function in sperm cell nuclear migration remains to be determined.

Along with the development of actin-based cellular dynamics in plants, the centrosome loss in somatic cells was established in bryophytes during land plant evolution (*Vaughn and Harper, 1998*). The diversification and expansion of actin nucleator *formin* genes are already evident in bryophytes (*Grunt et al., 2008*); it is still not yet clear how *ROP* signaling pathway genes in plants initiated and diversified during land plant evolution. However, several higher land plant features such as pollen tube tip growth (*Guan et al., 2013*), secondary cell wall formation (*Oda and Fukuda, 2012*), pavement cell indentation (*Xu et al., 2010*), and actin-based sperm cell nuclear migration described here are controlled by specific *ROP* signaling pathways. These observations suggest that extended roles of actin cytoskeleton at least by ROP signaling pathways through land plant evolution might have paved the way to the complete loss of microtubule–centrosome-dependent processes in flowering plants together with a dramatic impact on plant cell biology including reproduction mechanisms, cell divisions and cell structures.

## Materials and methods

### Plant material and growth conditions

All *Arabidopsis* plant lines used in this study are Columbia-0 (Col) ecotype, except the *porcino (por)* mutant in Landsberg *erecta* (Ler). Seeds were first germinated under short day conditions at 16°C (8 hr light and 16 hr dark) and kept for 1–2 weeks. Plants were then shifted to long day conditions at 20°C (16 hr light and 8 hr dark). The *pH3.10::H3.10-mRFP1* (*Ingouff et al., 2007*) and *pEC1::H2B-mRFP1* (*Ingouff et al., 2009*) lines have been described previously. The *p35S::sGFP-ROP8* and *pAGL80::tagRFP-TUA5* transgenic lines are gifts from Dr Zhenbiao Yang at UC Riverside, USA and Dr Shuh-ichi Nishikawa at Niigata University, Japan, respectively.

### Plasmid construction and transformation

All fragments were amplified by PCR using the KOD-plus ver. 2 PCR kit (TOYOBO, Japan). Primer sequences for PCR are listed in *Supplementary file 1A* and all constructs (*Supplementary file 1B*) were generated by the Multisite Gateway Technology (Invitrogen, CA, USA). The Lifeact-Venus, the omega translational enhancer Ω (*Gallie, 2002*), Ω-Venus, Ω-H2B, the dominant-negative form of *ACTIN 8* (*DN-ACTIN*), the constitutively-active and dominant-negative forms of *ROP8(CA-ROP8* and *DN-ROP8;Yang, 2002*), NLS-mCherry, and the ribosomal skip peptide (*El Amrani et al., 2004*; *Kim et al., 2011*) P2A-DN-ACTIN fragments were generated by Overlap Extension PCR (*Horton, 1993*).

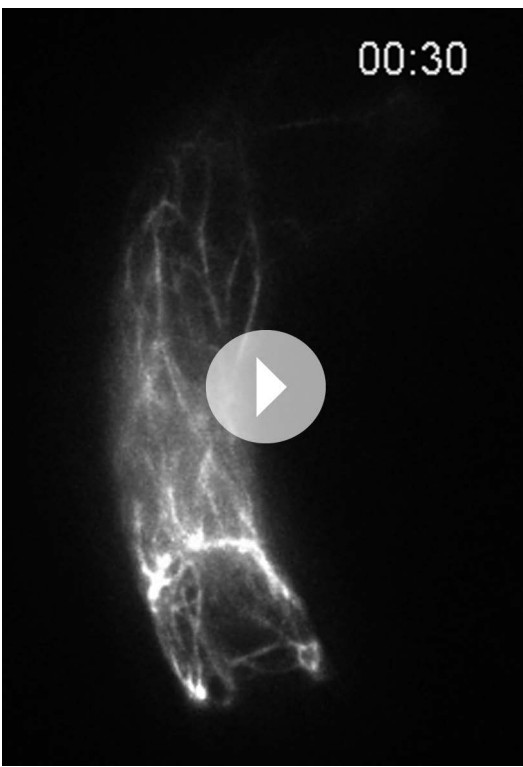

**Video 6**. Time-lapse images of F-actin dynamics in the mature WT central cell in the presence of the myosin inhibitor BDM (50 mM). F-actin in the central cell (white) is marked by *pFWA::Lifeact-Venus*. Images were taken at 30-s intervals.

The multisite gateway binary vector pAlligatorG43 was generated by replacing the *EN35S* expression cassette of pAlligator2 (*Bensmihen et al., 2004*) with the multisite gateway cassette. The GFP selection marker gene of pAlligatorG43 was replaced with mCherry to generate pAlligatorR43. Other multiple gateway vectors, pK7m24 GW,3, pH7m24 GW,3, pB7m24 GW,3, have been described previously (*Karimi et al., 2007*). All constructs were transformed into Col except *por* mutant (Ler) using the floral dip method (*Clough and Bent, 1998*).

## Microscopy settings and image processing

Fluorescence was acquired with laser-scanning confocal microscopy (63×/1.3 glycerol immersion objective lens; Leica SP5, Germany) for Venus (excitation 514 nm and emission BP 520–585 nm) and for mRFP1 and mCherry (excitation 594 nm and emission BP 600–650 nm), and a microscope (40×/1.30 and 60×/1.40 oil immersion objective lens; binning of 2; Nikon Ti-E, Japan) equipped with a disk-scan confocal system (Yokogawa CSU-X1, Japan) and CCD and CMOS camera (Photometrics CoolSNAP-HQ2, AZ, USA; Hamamatsu Photonics ORCA-Flash4.0, Japan). Time-lapse images were acquired every 0.5–1 min using multiple z-planes (7–10 planes, 1–1.5 µm intervals, objective piezo drive system) with the spinning disk confocal microscopy. Digital image processing was performed with LAS AF Lite (Leica, Germany), Metamorph (Molecular Devices, CA, USA), ImageJ (http://rsbweb.nih.gov/ij/), and/or Adobe Photoshop (Adobe Systems, CA, USA).

## Dexamethasone induction experiment

The experimental strategy was modified from previously described (*Samalova et al., 2005*). Mature emasculated pistils (36 hr after emasculation), harboring *pFWA::Lifeact-Venus,pFWA::LhGR*, and *pOPx4::NLS-mCherry-P2A:DN-ACTIN*, were submerged into the DEX working solution (DEX [D1756; Sigma-Aldrich, MO, USA], 40 µM; Silwet L-77 [VIS-02; LEHLE Seeds, TX, USA], 0.0001% in ddH$_2$O). Pollens, harbouring *pH3.10::H3.10:Clover*, were pollinated 12 hr after the DEX induction. 8–10 hr after pollination, ovules were dissected and fertilization phenotypes were observed using confocal microscopy.

## Semi in vitro fertilization assay

The method was modified from previously described (*Hamamura et al., 2011*). In brief, pollen growth medium (14% sucrose, 0.001% boric acid, 1.27 mM Ca(NO$_3$)$_2$, 0.4 mM MgSO$_4$, 1.5% low gelling temperature agarose [A9414; Simga-Aldrich, MO, USA], pH 7.0 with KOH) was poured into a well of a glass bottom dish (D141410; Matsunami Glass IND., LTD., Japan) to make a thin layer. Stigmas from emasculated pistils were cut with a 27-gauge needle (Terumo, Japan) at the junction between the style and ovary and placed horizontally on the pollen growth medium, followed by hand-pollination with pollen from the *pH3.10::H3.10-mRFP1*. After 3-hr incubation at 22°C in the dark, fresh ovules were dissected from the *pFWA::Lifeact-Venus* and were placed in front of growing pollen tubes on the medium and incubated further 2 hr. The spinning disk confocal microscopy was used to generate time-lapse images of double fertilization.

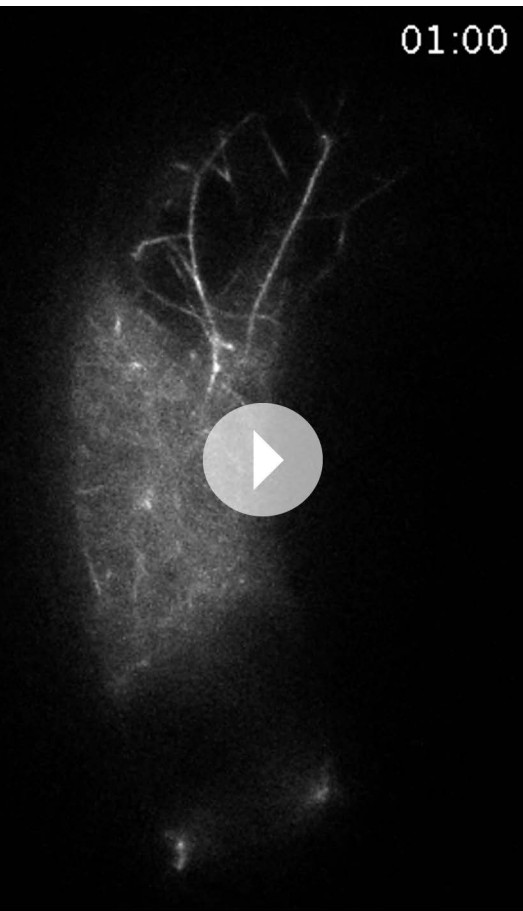

**Video 7.** Time-lapse images of F-actin dynamics in the mature WT central cell in the presence of the myosin inhibitor NEM (0.35 mM). F-actin in the central cell (white) is marked by *pFWA::Lifeact-Venus*. Images were taken at 30-s intervals.

## Quantification of F-actin dynamics

Particle image velocimetry (PIV) of the F-actin dynamics was performed on MATLAB using the PIVlab package (version 1.32) developed by William Thielicke and Eize J Stamhuis. PIV analysis was performed on the central cells as follows.

We first defined the region of interest (ROI) and overlaid a mask representing the shape of the central cell using the first image of the Video to restrict analysis only to the cell. Sizes of interrogation areas were optimized in order to calculate frame to frame displacement of F-actin. Higher spatial resolution was achieved by employing multi-pass window deformation technique, with a final size of 16 × 16 pixels (1.7 × 1.7 μm) and 50% overlap.

Once the velocity vector field was obtained, we then analyzed the vectors in two different ways. To analyze the F-actin movement towards or away from the nucleus, we restricted our analysis to around the nucleus. The component of velocity vectors around the nucleus in the direction of the center of nucleus was calculated and categorized into inward (towards the nucleus) and outward (away from the nucleus) velocity. The probability distribution of inward and outward velocities integrated over space and time was plotted as shown in *Figure 6B*.

To compare the average F-actin dynamics in the central cell of different genetic backgrounds and in the presence of myosin inhibitor BDM or NEM, the mean velocity of F-actin was measured. The probability distributions of velocities integrated over the entire central cell area and time were plotted as *Figure 6C*. The probability distribution of the velocity *f(v)* is then fitted by the gamma distribution (solid line in *Figure 6C*).

$$f(x;k,\theta) = \frac{x^{k-1}e^{-\frac{x}{\theta}}}{\Gamma(k)\theta^k}; x \geq 0 \, and \, k, \theta > 0$$

Where *k* is a shape parameter, *θ* is a scale parameter, and Γ(*k*) is the gamma function evaluated at *k*.

$$\Gamma(k) = \int_0^\infty x^{k-1}e^{-x}dx$$

The mean velocity of the probability distribution is represented by *kθ*. The summary of the mean velocity of different genetic backgrounds and in the presence of myosin inhibitor BDM or NEM was shown in *Figure 6D*. The error bars shown in *Figure 6D* are SEM. The p-values are calculated using student's t-test.

## F-actin dynamics assay

Ovules 2 days after emasculation were dissected and submerged into the assay medium (5% sucrose, half strength Murashige and Skoog salt, pH 5.7) in a glass bottom dish (D141410; Matsunami Glass IND. LTD., Japan). LatA (L5163; Sigma-Aldrich, MO, USA; 10 mM in DMSO), BDM (B0753; Sigma-Aldrich, MO, USA; 500 mM in the assay buffer), and NEM (E1271; Sigma-Aldrich, MO, USA; 500 mM in ethanol) were supplemented to generate fresh 100 μM LatA, 50 mM BDM, and 0.35 mM NEM assay media, respectively. The spinning disk confocal microscopy was used to generate time-lapse images of F-actin dynamics.

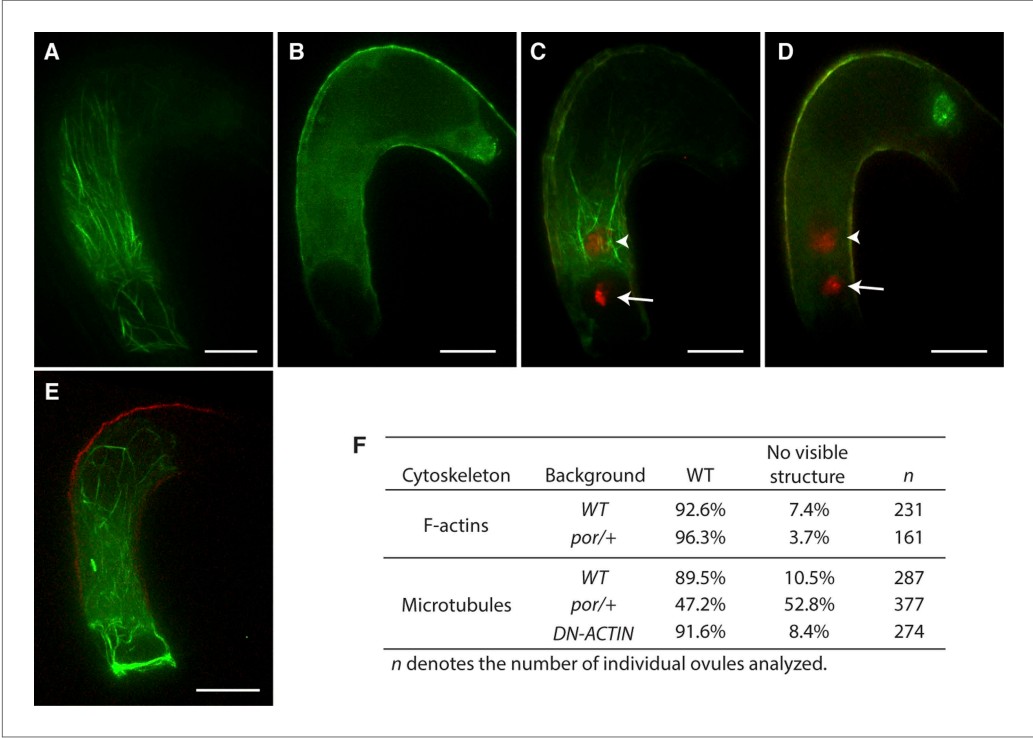

**Figure 7**. Effect of microtubule disruption on cytoskeleton in the central cell. (**A** and **B**) Microtubule structures (green) in the mature WT (**A**) and the *tubulin folding cofactors c (por)/+* heterozygous mutant (**B**) central cells marked by TagRFP-TUA5 under the control of the *AGL80* promoter. (**C** and **D**) In *por/+* plants, fertilization is successful in all ovules even though they do not display organized microtubules (**D**) like WT (**C**). Arrows and arrowheads point to the sperm cell chromatin (red), already decondensed in the egg cell and central cell nucleus, respectively. (**E**) F-actin in *por/+* heterozygous mutant visualized by the Lifeact-Venus under the control of the *FWA* promoter. (**F**) Percentages of ovules that show cytoskeleton structures in the mature central cell of different genetic backgrounds. Scale bar = 10 μm.

## GeneChip data analysis

Affymetrix *Arabidopsis* ATH1 22k GeneChip raw data of the female gametophyte cells (*Wuest et al., 2010*), young seed compartments (*Belmonte et al., 2013*), adult tissues (*Le et al., 2010*), and male gametophyte cells (*Borges et al., 2008*) were obtained from ArrayExpress in EBI (https://www.ebi.ac.uk/arrayexpress/) and Gene Expression Omnibus in NCBI (http://www.ncbi.nlm.nih.gov/geo/) websites. The raw data were processed and normalized using GCRMA in R. Hierarchical clustering analysis on ROP genes expression was carried out using CLC Main Workbench 7.02 (CLC bio, Denmark).

## Acknowledgements

We thank Z Yang (*p35S::sGFP-ROP8*) and S Nishikawa (*pAGL80::tagRFP-TUA5*) for seeds and M Mishra for his critical comments on our manuscript. This work was supported by Temasek Life Sciences Laboratory (to TK, DM, JL, RY, YT, and FB), National University of Singapore (to MS, YT, and FB), the Mechanobiology Institute at National University of Singapore (to YT), and the Japan Society for the Promotion of Science Fellowships (grant 6526 to DM). Author contributions are listed in the supplementary materials.

## Additional information

### Funding

| Funder | Grant reference number | Author |
| --- | --- | --- |
| Temasek Life Sciences Laboratory | | Tomokazu Kawashima, Daisuke Maruyama, Jing Li, Ramesh Yelagandula, Yusuke Toyama, Frédéric Berger |

| Funder | Grant reference number | Author |
|---|---|---|
| National University of Singapore | Department of Biological Sciences | Murat Shagirov, Yusuke Toyama |
| National University of Singapore | Mechanobiology Institute | Yusuke Toyama |
| Japan Society for the Promotion of Science | 6526 | Daisuke Maruyama |
| Nagoya University | | Daisuke Maruyama, Yuki Hamamura |

The funders had no role in study design, data collection and interpretation, or the decision to submit the work for publication.

## Author contributions

TK, Conception and design, Acquisition of data, Analysis and interpretation of data, Drafting or revising the article; DM, YH, Performed most live-cell imaging analyses presented; MS, YT, Performed F-actin dynamics analyses; JL, Generated the multisite gateway vector constructs; RY, Performed some confocal image analyses; FB, Conception and design, Analysis and interpretation of data, Drafting or revising the article

## Author ORCIDs

Tomokazu Kawashima, http://orcid.org/0000-0003-3803-3070
Jing Li, http://orcid.org/0000-0002-0989-7622

## Additional files

**Supplementary file**
• Supplementary file 1. (**A**) The list of primer sequences used for construct generation. (**B**). The list of Gateway vector constructs generated in this work.

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
