## [Decision Letter]

Thank you for sending your work entitled “The dynamic F-actin movement is essential in fertilization of flowering plants” for consideration at *eLife*. Your article has been favorably evaluated by Detlef Weigel (Senior editor) and 3 reviewers, one of whom, Sheila McCormick, is a member of our Board of Reviewing Editors.

The Reviewing editor and the other reviewers discussed their comments before we reached this decision, and the Reviewing editor has assembled the following comments to help you prepare a revised submission.

This manuscript reports a series of sound and creative experiments aimed at understanding how the sperm nucleus migrates to the female nucleus during fertilization in flowering plants. There is currently no mechanistic insight into this process in intact plants and thus this is significant work that impacts our understanding of plant reproduction in an important way. The central conclusions are backed up with multiple experimental approaches. The chief innovation of this study was the use of cell-type specific promoters to drive dominant negative ACTIN (egg and central cell) and ROP8 variants (central cell only). The images of the actin cytoskeleton in these target cells are stunning. Although Ohnishi et al used an artificial fusion system with isolated rice gametes to show an aster-like actin structure in the egg around the sperm nucleus, and that lat B disrupts sperm nuclear migration, they lacked genetic support for their pharmacological data.

The reviewers felt nevertheless that some of the comparisons of plants to animals should be tempered. Furthermore, additional background information will make the manuscript more accessible to the general reader. We have enumerated below the issues that should be addressed in the revision.

1) All three reviewers question the importance of the figure supplements devoted to evolution of ROPs and associated pathways; they are not described in the results and it is difficult to determine whether there has been a significant diversification and expansion of these gene families relative to others. These figures are also poor quality relative to the rest of the paper. Is the rate of diversification significantly different from other gene families? Finally, we are not convinced that this analysis is sufficient to make the claim in the abstract that “Expansion and diversification of actin regulators during land plant evolution might have not only enabled the innovation of an actin-based mechanism of fertilization…” These results, if retained in the revised manuscript, need to be substantially improved and incorporated into the Results section, but we think they can equally well be omitted with no loss of impact to the major findings of this manuscript.

2) *eLife* is an online journal with no space limitations “Several independent *DN-ACTIN* transgenic lines were investigated and showed similar defect rates (data not shown)”: “data not shown” is not necessary; show the results from the other lines; “30% defects in *DN-ACTIN* (n=120)…” how many independent lines, as per comment about DN-ACTIN in egg; “Occasionally, a few nuclei divisions were observed in the endosperm (data not shown)”: why data not shown? What is occasionally? 2% 10%?Please show the data; “only *DN-ROP8* lines showed fertilization defects similar to those observed in central cells expressing DN-ACTIN”: lines are mentioned; how many and what was the range of the defective fertilization?

3) The authors have convincingly shown that actin is required for sperm nuclear migration in the egg and central cell. However, they have not shown that this is due to direct attachment of sperm to actin via myosin. The (unlikely) possibility remains that an intact actin cytoskeleton is necessary for sperm nuclear migration, but this requirement is indirect rather than direct. The authors address this possibility with a very elegant experiment that uses an inducible promoter system to transiently express a dominant negative actin after central cell development is complete. This experiment rules out the possibility that development of the central cell has been altered due to prolonged cytoskeletal disruption, however, it does not rule out the possibility that the defects in sperm nuclear migration are due to general perturbation of cellular structure rather than a direct effect on actin-myosin-based sperm nuclear migration. The discussion of this point should be refined.

4) ROP8 is specific to the central cell. The authors only address the function of ROP8 in the central cell and conclude that ROP8 is required for sperm cell migration in the central cell. The introduction makes it seem as though ROP8 controls actin dynamics in both female gametes. The authors should make it clear in the discussion and introduction that they have uncovered a potential component of the mechanism that regulates actin dynamics in the central cell, but this mechanism remains unknown for the egg.

5) The formation of aster-shaped actin structures around sperm nuclei is the best evidence (along with pharmacological disruption by myosin inhibitors) presented for a direct association between actin and sperm nuclei. This section could be improved by clarifying whether asters form in the absence of sperm nuclei and by quantifying the rate the of observed aster formation in the presence of sperm nuclei.

6) It is well known that ROPs are associated with plasma membranes, so the root epidermal experiment is not necessary. Add to the previous sentences, e.g. “as expected for a ROP”.

---

## [Author Response]

*1) All three reviewers question the importance of the figure supplements devoted to evolution of ROPs and associated pathways; they are not described in the results and it is difficult to determine whether there has been a significant diversification and expansion of these gene families relative to others. These figures are also poor quality relative to the rest of the paper. Is the rate of diversification significantly different from other gene families? Finally, we are not convinced that this analysis is sufficient to make the claim in the abstract that “Expansion and diversification of actin regulators during land plant evolution might have not only enabled the innovation of an actin-based mechanism of fertilization…” These results, if retained in the revised manuscript, need to be substantially improved and incorporated into the Results section, but we think they can equally well be omitted with no loss of impact to the major findings of this manuscript*.

We agree that the phylogenetic study of actin regulators should be extended if we would consider including it in the “results” and thus have omitted this evolutional discussion and its mention in the Abstract.

2) eLife is an online journal with no space limitations “Several independent DN-ACTIN transgenic lines were investigated and showed similar defect rates (data not shown)”: “data not shown” is not necessary; show the results from the other lines; “30% defects in DN-ACTIN (n=120)…” how many independent lines, as per comment about DN-ACTIN in egg; “Occasionally, a few nuclei divisions were observed in the endosperm (data not shown)”: why data not shown? What is occasionally? 2% 10%? Please show the data; “only DN-ROP8 lines showed fertilization defects similar to those observed in central cells expressing DN-ACTIN”: lines are mentioned; how many and what was the range of the defective fertilization?

We added results of other transgenic lines and detail numbers of observations in the text accordingly.

*3) The authors have convincingly shown that actin is required for sperm nuclear migration in the egg and central cell. However, they have not shown that this is due to direct attachment of sperm to actin via myosin. The (unlikely) possibility remains that an intact actin cytoskeleton is necessary for sperm nuclear migration, but this requirement is indirect rather than direct. The authors address this possibility with a very elegant experiment that uses an inducible promoter system to transiently express a dominant negative actin after central cell development is complete. This experiment rules out the possibility that development of the central cell has been altered due to prolonged cytoskeletal disruption, however, it does not rule out the possibility that the defects in sperm nuclear migration are due to general perturbation of cellular structure rather than a direct effect on actin-myosin-based sperm nuclear migration. The discussion of this point should be refined*.

We agree that it is impossible to rule out that the effect of actin disruptions is indirect. Yet, we could not detect any changes of cell structure beside actin filaments in DN-ACTIN lines although these might be below the obvious detection. Therefore, we added a sentence of caution in the Discussion as follows:

“However, it still remains to be determined whether F-actin directly interacts with the sperm cell nucleus for migration. Further investigations should be made to rule out the possibility that the sperm cell nuclear migration defect by DN-ACTIN expression is not due to general perturbation of cellular structure.”

*4) ROP8 is specific to the central cell. The authors only address the function of ROP8 in the central cell and conclude that ROP8 is required for sperm cell migration in the central cell. The introduction makes it seem as though ROP8 controls actin dynamics in both female gametes. The authors should make it clear in the discussion and introduction that they have uncovered a potential component of the mechanism that regulates actin dynamics in the central cell, but this mechanism remains unknown for the egg*.

The problem resulted from our attempt to avoid botanical terms for animal biologists. We have changed this and specified “central cell” and “egg cell” throughout where it was required. We also added the following statement to clarify that the mechanism in the egg cell remains to be determined: “It is plausible that a distinct ROP plays a similar role during egg cell fertilization, but it is also possible that other mechanisms control F-actin-dependent migration of the male nucleus to the egg cell nucleus.”

*5) The formation of aster-shaped actin structures around sperm nuclei is the best evidence (along with pharmacological disruption by myosin inhibitors) presented for a direct association between actin and sperm nuclei. This section could be improved by clarifying whether asters form in the absence of sperm nuclei and by quantifying the rate the of observed aster formation in the presence of sperm nuclei*.

We added the following sentence in the result section to clarify our aster-like structure observation: “This aster-like structure of F-actin was only observed around the sperm cell nucleus in the central cell after plasmogamy (100%, n=48) and was never observed in unfertilized central cells (0%, n=72). Consistently, the time-lapse imaging showed the presence of aster-like F-actin structure only during sperm cell nucleus migration [100% (n=5); 0% in unfertilized ovule (n=14)].”

*6) It is well known that ROPs are associated with plasma membranes, so the root epidermal experiment is not necessary. Add to the previous sentences, e.g. “as expected for a ROP”*.

Even if ROPs are generally associated with the plasma membrane, we show that ROP8 controls novel F-actin dynamics in the gametophytic central cell. Therefore, we think that it is of importance to test as a control that it does localize to the plasma membrane in somatic cell. To make this point clear, we added the following sentences in the result and discussion section, respectively: “indicating that ROP8 sub-localization is controlled by a general machinery common between the gametophytic central cell and somatic root epidermal cell” and “Although the machinery by which ROP8 is associated to the plasma membrane is general and the activity of ROP8 in the central cell should be investigated further, this putative constant active mode of ROP8 in the central cell might shed light on a new facet of Rho-GTPase function in plant cells.”